# G²-Occ: Geometry-Guided Gaussian Primitives for Embodied Semantic Occupancy Prediction

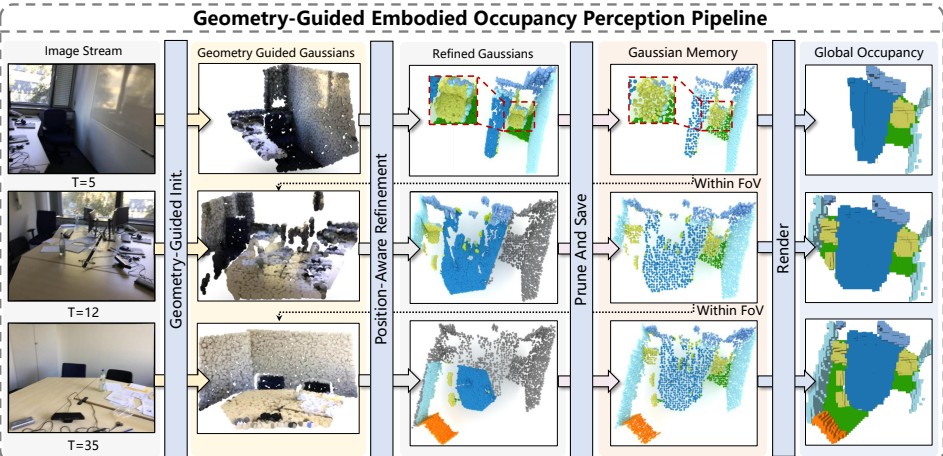

Figure 1: Illustration of the Geometry-Guided Embodied Occupancy Perception Pipeline.

## Abstract

This paper introduces a vision-only framework for embodied semantic occupancy prediction based on geometry-guided Gaussian primitives. Our approach implicitly recover scene geometry from monocular color images via pre-trained depth and normal estimation models. The core of our framework departs from traditional random or uniform initialization strategies, instead leveraging the recovered geometric priors to effectively manage the entire lifecycle of Gaussian primitives, including initialization, updating and eventual pruning. Specifically, we design a Geometry-Guided Initialization module that utilizes the recovered geometry to generate Gaussian primitives within potentially occupied regions of the scene, ensuring a rational and efficient primitive distribution from the outset. Subsequently, we propose a Position-Aware Scene Update and Pruning pipeline, which integrates a Position-Aware Gaussian Refinement process and Confidence-Based Fusion and Pruning module. This pipeline is responsible for maintaining the global consistency of the scene representation across continuous online observations while adaptively fusing redundant primitives to manage computational complexity. The effectiveness and advanced nature of our method are thoroughly validated through extensive experiments on four popular indoor semantic occupancy prediction benchmarks, where it demonstrates state-of-the-art performance.

## 1 Introduction

3D scene understanding is a cornerstone technology for frontier domains such as embodied AIZhang et al. (2025a); Liu et al. (2024b); Wang et al. (2024) and autonomous drivingHuang et al. (2024a); Humblot-Renaux et al. (2022); Tian et al. (2025); Ji et al. (2025). A key task in this area is 3D semantic occupancy prediction, which aims to jointly infer a scene's geometry and semantics from visual inputs, providing a dense, holistic representation for downstream tasks like robot navigation.

Existing approaches to this task largely fall into two paradigms. Projection-based methods Li et al. (2022); Huang et al. (2021; 2023) represent 3D information on 2D planes (*e.g.*, BEV or TPV),

striking a balance between accuracy and efficiency. In contrast, dense voxel-based methods Cao & De Charette (2022); Wei et al. (2023); Ma et al. (2024) employ feature lifting mechanisms to learn explicit 3D representations, achieving high-fidelity results. However, the dense voxel paradigm suffers from a critical scalability issue. The computational and storage overhead grows cubically with resolution, resulting significant resources wasted on empty voxels. This severely limits its applicability in high-resolution scenarios and motivates the need for more efficient representations.

Leveraging the universal approximation capabilities of GMMs Dalal & Hall (1983); Goodfellow et al. (2016), a new paradigm using sparse 3D Gaussian primitives has emerged to address the limitations of dense representations. The pioneering work, GaussianFormer Huang et al. (2024b), demonstrated that by adaptively encoding local scene geometry and semantics, this approach circumvents the computational overhead of voxels while achieving state-of-the-art performance.

This paradigm was subsequently extended to the embodied setting by EmbodiedOcc Wu et al. (2024), which introduced a memory bank to incrementally update a uniformly initialized set of primitives from a sequential video feed. While this line of work validates the approach's effectiveness, it exposes a critical challenge that we address: **How can we design a comprehensive lifecycle management strategy for Gaussian primitives that ensures efficient initialization, globally consistent updates, and effective complexity control in embodied settings?**

Lacking effective geometric guidance, prevailing methods commonly adopt random or uniform initialization strategies, which indiscriminately scatter Gaussian primitives throughout the 3D space. The fundamental flaw in this approach is its disregard for the inherent non-uniformity of geometric and semantic distributions in real-world scenes. Specifically, this strategy inevitably leads to a sub-optimal allocation of primitives. It results in underfitting in regions rich with geometric details, leading to insufficient expressive capacity, while causing over-sampling and significant redundancy in structurally simple areas, thereby wasting valuable computational and memory resources.

To address the limitations of prior work, we introduce a novel framework for geometry-guided embodied occupancy perception as shown in Fig. 1. The central principle of our framework is to discard inefficient uniform initialization strategies and instead leverage geometric priors recovered from monocular images to guide the entire lifecycle of Gaussian primitives, including their initialization, updates, and pruning.

To this end, the framework integrates two key innovative components: (1) A geometry-guided initialization module that utilizes estimated depth and normal map as geometric priors to adaptively generate Gaussian primitives on potentially occupied regions of the scene. This approach addresses the sub-optimal primitive allocation problem from the outset. (2) A Position-Aware Scene Update and Pruning pipeline which is composed of a Position-Aware Gaussian Refinement process and a Confidence-Based Fusion and Pruning module. The former ensures perceptual consistency across continuous observations by injecting global positional encodings, while the latter introduces a confidence metric to adaptively fuse and prune redundant primitives.

The main contributions of this work are threefold:

- We propose a Geometry-Guided Initialization module for Gaussian primitives. By leveraging geometric priors recovered from monocular images, this method significantly enhances both the efficiency of the initialization process and the performance on downstream tasks.

- We propose a Position-Aware Scene Update and Pruning pipeline which integrates a Position-Aware Gaussian Refinement process to ensure spatiotemporal consistency and a Confidence-Based Fusion and Pruning module to efficiently prune redundant primitives.

- We conduct extensive experiments on four mainstream indoor 3D semantic occupancy prediction benchmarks, demonstrating that our method establishes a new state-of-the-art, significantly outperforming prior approaches.

## 2 RELATED WORK

**3D Semantic Occupancy Prediction.** This task jointly infers scene geometry and semantics. While early methods Song et al. (2017); Peng et al. (2020); Chen et al. (2020) leveraged explicit 3D inputs like depth or TSDFs for strong geometric priors, MonoScene Cao & De Charette (2022) pioneered a vision-only approach. However, this monocular paradigm introduced inherent depth ambiguity. Subsequent research has largely focused on mitigating this ambiguity. One line of work integrates

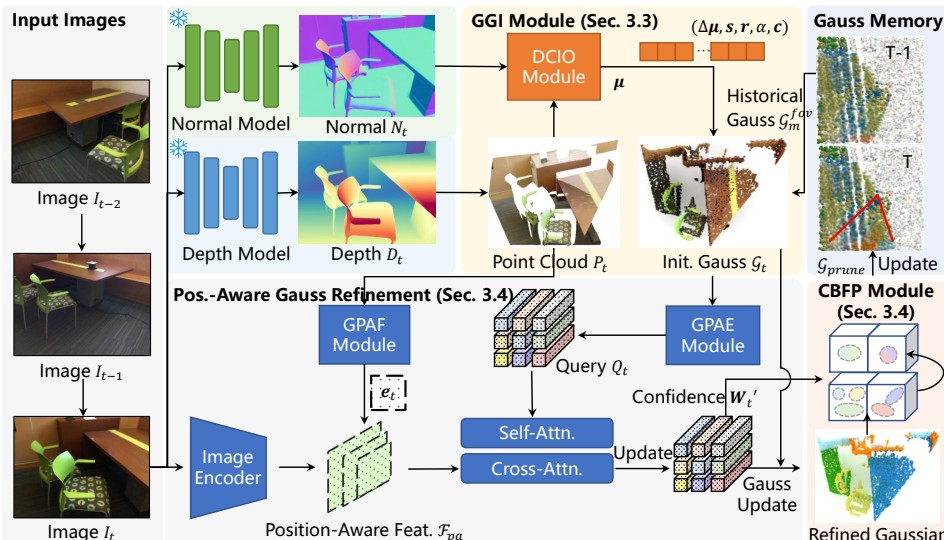

Figure 2: Overall architecture of the proposed method.

explicit depth priors from stereoShamsafar et al. (2022) or pre-trained estimators into U-Net-based architectures Miao et al. (2023); Yao et al. (2023); Yu et al. (2024); Liu et al. (2024a); Tong et al. (2023). Another line of work employs query-based decoders, using depth information to guide the initial placement of queries for more effective scene representation Zhang et al. (2023); Jiang et al. (2024); Lu et al. (2024). Despite their progress, these methods, which typically rely on dense voxel grids, are fundamentally limited by their computational overhead. A significant portion of resources is consumed by processing empty grids, which hinders their scalability to higher resolutions.

**Gaussian-based Scene Representation.** To overcome the inefficiency of dense voxel grids, recent works Huang et al. (2024b; 2025); Zhu et al. (2025); Boeder et al. (2025); Zuo et al. (2025); Chambon et al. (2025) have explored sparse, explicit representations, with 3D Gaussian primitives emerging as a promising direction. GaussianFormer pioneered this approach by rendering a sparse set of Gaussian primitives into a dense grid. This paradigm was subsequently adapted to the online, embodied setting by EmbodiedOcc Wu et al. (2024) and its follow-ups Zhang et al. (2025b); Wang et al. (2025a). However, these methods inherit a critical limitation: they rely on a naive random or uniform initialization strategy. This approach disregards the inherent non-uniformity of real-world scenes, leading to a sub-optimal allocation of primitives. It results in under-sampling in geometrically complex regions and over-sampling in simple areas. In contrast, our method directly tackles this limitation by leveraging geometric priors to guide the entire lifecycle of the Gaussian primitives, from their initial placement to their subsequent refinement and pruning.

## 3 METHODOLOGY

### 3.1 PROBLEM FORMULATION

We address the task of embodied semantic occupancy prediction, where an agent incrementally builds a global 3D scene representation from a sequence of monocular RGB images. At each timestep $t$, the scene state is represented by a set of 3D Gaussian primitives $\mathcal{G}_{t-1}$. Given a new image $\boldsymbol{I} \in \mathbb{R}^{H \times W \times 3}$ and its corresponding camera parameters $(\boldsymbol{K}_t, \boldsymbol{E}_t)$, our model $\mathcal{M}$ recurrently updates the scene state to produce the new representation $(\mathcal{V}_t, \mathcal{G}_t)$:

$$(\mathcal{V}_t, \mathcal{G}_t) = \mathcal{M}(\boldsymbol{I}_t, \boldsymbol{K}_t, \boldsymbol{E}_t, \mathcal{G}_{t-1}, \mathcal{V}_{t-1}), \tag{1}$$

where $\mathcal{V}_t \in \{0, 1, ..., C-1\}^{X \times Y \times Z}$ is a grid of semantic labels, with class 0 denoting empty space. The scene representation $\mathcal{G}_t$ is a set of 3D Gaussian primitives, where each primitive is defined by its position $\boldsymbol{\mu}_i \in \mathbb{R}^3$, scale $\boldsymbol{s}_i \in \mathbb{R}^3$, rotation quaternion $\boldsymbol{q}_i \in \mathbb{R}^4$, opacity $\alpha_i \in [0, 1]$, and semantic logits $\boldsymbol{c}_i \in \mathbb{R}^C$. The single-frame prediction, or local occupancy prediction, can be seen as a special case of this formulation where the initial state $\mathcal{G}_{t-1}$ and $\mathcal{V}_{t-1}$ are empty sets.

## 3.2 OVERALL ARCHITECTURE

We propose the *Geometry-Guided Embodied Occupancy Perception Pipeline* for both local and embodied occupancy prediction, as illustrated in Fig. 2. Generally, this pipeline is designed to preferentially initialize Gaussian primitives in potentially occupied regions of the scene based on geometric information. Afterwards, rigorously maintaining the global consistency of the entire scene representation. Finally, dynamicly fuse and prune redundant Gaussian primitives.

Specifically, given an input image $I_t$, we first employ a lightweight encoder to extract multi-scale features $\mathcal{F}_{rgb} = \{f_{rgb}^l\}_{l=1}^{L}$, where $L$ is the number of scale levels, and leverage pre-trained models to estimate a depth map $D_t$ and a normal map $N_t$. A key contribution of our work is the *Geometry-Guided Initialization* (GGI) module $\mathcal{M}_i$, which uses the geometric priors from $D_t$ and $N_t$ to adaptively initialize a set of new Gaussian primitives $\mathcal{G}_t^p$ in potentially occupied regions. This approach overcomes the limitations of the random or uniform initialization strategy used in prior works Wu et al. (2024). For each timestep, the newly generated primitives $\mathcal{G}_t^p$ are merged with historical primitives $\mathcal{G}_m^{fov}$ from the Gaussian memory that are within the current frustum. This combined set $\mathcal{G}_t$ is then refined by the *Position-Aware Gaussian Refinement* (PAGR) process. First, the *Global Position-Aware Feature Refinement* (GPAF) module and the *Global Position-Aware Encoder* (GPAE) module enrich the image features and corresponding primitive queries with absolute world-coordinate context. Subsequently, these features are refined through several cycles of self- and cross-attention. The refined primitives $\mathcal{G}_t'$ are then fused and pruned by our *Confidence-Based Fusion and Pruning (CBFP)* module $\mathcal{M}_p$. The final set of pruned primitives $\mathcal{G}_{prune}$ is then used to update the Gaussian memory for the next timestep and render the occupancy prediction.

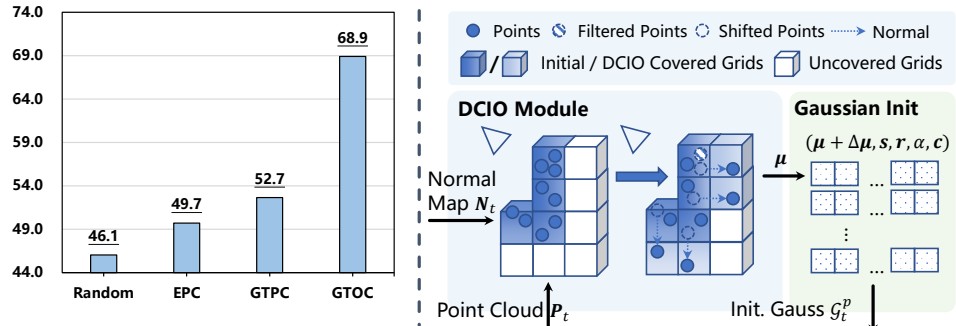

Figure 3: Illustration of the Geometry-Guided Initialization module.

## 3.3 GEOMETRY-GUIDED INITIALIZATION MODULE

Query-based approaches represent a highly effective pathway for semantic occupancy prediction. However, a core challenge for such methods lies in effectively guiding the queries to Regions of Interest (RoIs). This creates a circular dependency problem: queries must be pre-localized to target regions to extract meaningful features, yet this localization process itself relies on scene priors derived from those very features. Therefore, departing from prior works that randomly or uniformly initialize query anchors across the entire space, we propose a Geometry-Guided Initialization module. This module leverages camera intrinsics and extrinsics, along with estimated depth and normals, to precisely initialize Gaussian primitives in potentially occupied regions of the scene, thereby aiming to break the aforementioned circular dependency.

**Proxy Experiment.** Our work is motivated by the hypothesis that preferentially initializing Gaussian primitives in occupied regions is a more effective and efficient strategy than random or uniform initialization. To validate this, we conduct a proxy experiment, as shown in Fig. 3, comparing four initialization strategies while keeping the total primitive count and all other hyperparameters constant: (1) Random (baseline); (2) EPC, initializing on a point cloud from estimated depth, which mimics our method's input; (3) GTPC, using a point cloud from ground-truth depth to isolate initialization from depth errors; and (4) GTOC, initializing on ground-truth occupancy centers to establish a practical performance upper bound.

The results confirm our hypothesis, showing a clear performance ranking: GTOC > GTPC > EPC > Random. This demonstrates that performance scales directly with the quality of the geometric prior, providing strong empirical support for our geometry-guided approach.

Building on the findings from the proxy experiment, we first back-project the input depth map $\boldsymbol{D} = \{\boldsymbol{d}_h\}_{h=0}^{H \times W}$ into a dense pseudo-point cloud $\boldsymbol{P} = \{\boldsymbol{p}_h\}_{h=0}^{H \times W} = \{\boldsymbol{E} \cdot [(\boldsymbol{d}_h \cdot \boldsymbol{K}^{-1} \cdot \boldsymbol{x}_h)^T, 1]^T\}_{h=0}^{H \times W}$ using the standard pinhole camera model, where $\boldsymbol{x}_h = [v, u, 1]^T$ represents the homogeneous coordinates of the pixel $(u, v)$. For brevity, the timestamp subscript $t$ is omitted here. However, this initial point cloud is both computationally expensive due to its density and insufficient as it only covers object surfaces. To address this, we introduce a *Density Control and Internal Offset* (DCIO) module, which first performs voxel-based downsampling on $\boldsymbol{P}$ and then shifts the resulting primitives inwards along their estimated normals to ensure coverage of object interiors.

First, to manage density, we perform voxel-based downsampling on $\boldsymbol{P}$, where the point cloud is initially partitioned into a 3D grid $\boldsymbol{V}_s = \{v_s\}_{s=0}^{S} = \{\lfloor \boldsymbol{p}_h/\epsilon \rfloor\}_{s=0}^{S}$ based on a predefined voxel size $\epsilon$, where $S$ is the total number of non-empty voxels. Subsequently, at most $k = \lceil N_p/S \rceil$ points are randomly sampled within each non-empty voxel. Second, to ensure coverage of object interiors, we apply a normal-guided offsetting mechanism. For each sampled point $\boldsymbol{p}_i$, we estimate its normal vector $\boldsymbol{n}_i'$, transform it to the world coordinate system $\boldsymbol{n}_{w,i}$ and then offset the point along this direction:

$$\boldsymbol{N}_w = \{\boldsymbol{n}_{w,i}\}_{i=1}^{N_p} = \{\boldsymbol{E} \cdot [\boldsymbol{n}_i', 1]^T\}_{i=1}^{N_p}, \tag{2}$$

$$\boldsymbol{\mu} = \{\boldsymbol{\mu}_s\}_{s=0}^{S} = \{\boldsymbol{p}_i + \boldsymbol{n}_{w,i} \times a \times \epsilon | a \in [0, k_s), \boldsymbol{p}_i \in \boldsymbol{P}_s\}_{s=0}^{S}, \tag{3}$$

where $\boldsymbol{P}_s$ is the set of $k_s$ points randomly sampled within the voxel $v_s$ and $a$ is the local index. This strategy achieves two goals: it provides coverage for object interiors by "pushing" primitives inward, and the varied offset distances prevent primitive clustering on object surfaces, enhancing structural completeness.

To compensate for potential inaccuracies in the geometry derived from estimated depth maps, our module introduces learnable offsets for all Gaussian primitive attributes. This allows for an adaptive, scene-aware initialization. A learnable positional offset $\Delta\boldsymbol{\mu}_i$ is added to the geometrically-derived center $\boldsymbol{\mu}_i$ to produce the final initial mean. A complete 3D semantic Gaussian primitive $\boldsymbol{G}_i$ is thus defined by its full set of properties:

$$\mathcal{G} = \{\boldsymbol{G}_i\}_{i=1}^{N_p} = \{(\boldsymbol{\mu}_i + \Delta\boldsymbol{\mu}_i, \boldsymbol{s}_i, \boldsymbol{r}_i, \alpha_i, \boldsymbol{c}_i)\}_{i=1}^{N_p}. \tag{4}$$

### 3.4 POSITION-AWARE SCENE UPDATE AND PRUNING PIPELINE

While our Geometry-Guided Initialization provides a strong per-frame prior, the central challenge in embodied perception is maintaining the scene representation's integrity and efficiency over time. To this end, we introduce two synergistic modules that manage the lifecycle of Gaussian primitives.

First, we introduce *Position-Aware Guassian Refinement* (PAGR) process that enhances the standard update pipeline Wu et al. (2024) by injecting absolute world-coordinate information, which is critical for augmenting spatiotemporal consistency and mitigating drift. Specifically, a *Global Position-Aware Feature* (GPAF) module is proposed to ground all updates in a stable global frame of reference by injecting absolute world-coordinate positional encodings into the multi-scale image features. The process begins by generating a positional encoding tensor $\boldsymbol{e} \in \mathbb{R}^{H \times W \times D_q}$ from the world coordinates of the dense point cloud $\boldsymbol{P}$, which is obtained by back-projecting the depth map. This is achieved using the sinusoidal positional encoding scheme Vaswani et al. (2017), defined for the $x$-axis as:

$$PE(x, 2u) = \sin(\frac{x}{\tau^{2u/\left\lfloor \frac{D_q}{3} \right\rfloor}}), PE(x, 2u+1) = \cos(\frac{x}{\tau^{2u/\left\lfloor \frac{D_q}{3} \right\rfloor}}), \tag{5}$$

where $\tau$ is a predefined temperature hyperparameter. After concatenating the encodings from all three axes, the resulting tensor $\boldsymbol{e}$ is downsampled via bilinear interpolation and added element-wise to each scale of the multi-scale image features $\mathcal{F}_{rgb}$ to yield position-aware features $\mathcal{F}_{pa}$:

$$\mathcal{F}_{pa} = \{\boldsymbol{f}_{pa}^l\}_{l=1}^{L} = \{\boldsymbol{f}_{rgb}^l + Bilinear(\boldsymbol{e}, (H_l, W_l))\}_{l=1}^{L}. \tag{6}$$

Similarly, to overcome the limitations of the camera-centric queries used in prior work Wu et al. (2024), a *Global Position-Aware Encoder* (GPAE) module is proposed to enrich the initialization of Gaussian query features $\mathcal{Q}$ by incorporating a world-coordinate positional encoding $\boldsymbol{e}_a^i$. This ensures better alignment with $\mathcal{F}_{pa}$ and results in a globally-aware query:

$$\boldsymbol{Q}_{pa}^i = MLP(z_c^i, d_i) + MLP(\boldsymbol{G}_i) + \boldsymbol{e}_a^i, \tag{7}$$

Table 1: **Local prediction performance.** We compare our method on OccScannet-Mini and Occ-Scannet datasets against several baselines: MonoScene Cao & De Charette (2022), ISO Yu et al. (2024), GA-MonoSSC Li et al. (2025b), EmbodiedOcc Wu et al. (2024), RoboOcc Zhang et al. (2025b), EmbodiedOcc++ Wang et al. (2025a), and SplatSSC Qian et al. (2025). * denotes models re-trained with the SplatSSC training settings for a fair comparison.

| Method | Dataset | IoU | mIoU | ceiling | floor | wall | window | chair | bed | sofa | table | tvs | furniture | objects |
|---|---|---|---|---|---|---|---|---|---|---|---|---|---|---|
| MonoScene | | 41.90 | 25.90 | 17.00 | 46.20 | 23.90 | 12.70 | 27.00 | 29.10 | 34.80 | 29.10 | 9.70 | 34.50 | 20.40 |
| ISO | | 42.90 | 29.40 | 21.10 | 42.70 | 24.60 | 15.10 | 30.80 | 41.00 | 43.30 | 32.00 | 12.10 | 35.90 | 25.10 |
| EmbodiedOcc | Occ-Scannet-Mini | 53.80 | 46.40 | 29.10 | 48.70 | 42.30 | 38.70 | 42.00 | 62.70 | 60.60 | 48.20 | 33.80 | 58.00 | 46.50 |
| RoboOcc | | 57.25 | 47.71 | - | - | - | - | - | - | - | - | - | - | - |
| EmbodiedOcc++ | | 55.70 | 48.20 | 23.30 | 51.00 | 42.80 | 39.30 | 43.50 | **65.60** | **64.00** | 50.70 | **40.70** | 60.30 | 48.90 |
| Ours | | **57.67** | **49.72** | **33.70** | **54.40** | **45.50** | **39.60** | **45.70** | 64.90 | 63.20 | **51.80** | 37.40 | **60.30** | **50.60** |
| EmbodiedOcc* | | 57.85 | 49.40 | 26.60 | 54.40 | 46.40 | 42.70 | 44.70 | 64.90 | 65.30 | 50.80 | 35.00 | 62.50 | 50.10 |
| SplatSSC* | Occ-Scannet-Mini | 61.47 | 48.87 | 36.60 | 55.70 | 46.50 | 40.10 | 45.60 | 64.50 | 62.40 | 48.60 | 30.60 | 61.20 | 45.39 |
| Ours* | | **61.99** | **54.00** | **37.50** | **59.90** | **49.60** | 42.80 | **49.50** | **68.80** | 56.30 | 40.90 | | **65.10** | **55.40** |
| MonoScene | | 41.60 | 24.62 | 15.17 | 44.71 | 22.41 | 12.55 | 26.11 | 27.03 | 35.91 | 28.32 | 6.57 | 32.16 | 19.84 |
| ISO | | 42.16 | 28.71 | 19.88 | 41.88 | 22.37 | 16.98 | 29.09 | 42.43 | 42.00 | 29.60 | 10.62 | 36.36 | 24.61 |
| GA-MonoSSC | | 48.59 | 35.65 | 26.01 | 51.89 | 30.35 | 24.58 | 30.34 | 48.44 | 49.92 | 37.59 | 22.48 | 43.13 | 27.41 |
| EmbodiedOcc | Occ-Scannet | 53.95 | 45.48 | 40.90 | 50.80 | 41.90 | 33.00 | 41.20 | 55.20 | 61.90 | 43.80 | 35.40 | 53.50 | 42.90 |
| EmbodiedOcc++ | | 54.90 | 46.20 | 36.40 | 53.10 | 41.80 | 34.40 | 42.90 | **57.30** | **64.10** | 45.20 | 34.80 | 54.20 | 44.10 |
| RoboOcc | | 56.48 | 47.67 | 45.36 | 53.49 | 44.35 | 34.81 | 43.38 | 56.93 | 63.35 | 46.35 | 36.12 | **55.48** | 44.78 |
| Ours | | 56.48 | **48.23** | **46.50** | **54.20** | **44.40** | **35.80** | **43.70** | 57.20 | 63.50 | **47.00** | **36.90** | 55.20 | **46.10** |
| EmbodiedOcc* | | 57.49 | 49.30 | 44.40 | 54.70 | 46.30 | 38.50 | 44.60 | 58.70 | 65.10 | 47.70 | 37.50 | 58.10 | 46.70 |
| SplatSSC* | Occ-Scannet | **62.83** | 51.83 | 49.10 | 59.00 | 48.30 | 38.80 | 47.40 | 62.40 | 67.00 | 49.50 | **42.60** | 60.70 | 45.40 |
| Ours* | | 60.43 | **52.69** | **52.40** | **59.30** | **48.90** | **40.30** | 47.10 | 60.70 | **67.40** | **51.00** | 42.20 | 60.50 | **50.00** |

where $e_a^i$ is the encoding of the primitive's world-coordinate position, while $z_c^i$ and $d_i$ are camera-frame depth features.

To manage memory complexity and prune redundant Gaussian primitives, we introduce the *Confidence-Based Fusion and Pruning* (CBFP) module. This module operates on the principle that spatially proximal primitives often represent the same object surface and can be merged without significant loss of accuracy. The process, illustrated in Fig. 2, begins by partitioning the primitives $\mathcal{G}_t^{'}$ into voxel-based clusters $\boldsymbol{V}_\psi = \{\lfloor \boldsymbol{\mu}/\psi \rfloor\}_{r=0}^{N_r}$. Crucially, instead of uniform averaging, we predict a confidence score for each primitive from its corresponding query feature $\mathcal{Q}^{'}$. These scores are then normalized via Softmax within each cluster to serve as fusion weights $\boldsymbol{W}'$:

$$\boldsymbol{W} = \sigma(\mathcal{M}_{conf}(\mathcal{Q}^{'})), \boldsymbol{W}' = \frac{\exp(W_i)}{\sum_{r:\boldsymbol{V}_\psi^i=r} \exp(W_r)}. \tag{8}$$

Finally, all primitives within a cluster are fused into a single new primitive by taking a confidence-weighted average of their attributes which is formulated as $\bar{a}_r = \sum_{r:\boldsymbol{V}_\psi^i=r} w_i a_i, a \in \{\boldsymbol{\mu}, \hat{\boldsymbol{r}}, \boldsymbol{s}, \alpha, \boldsymbol{c}\}$, where $\hat{\boldsymbol{r}}_i$ represents the normalized rotation quaternion. This confidence-driven mechanism allows the model to adaptively prune redundancy, effectively controlling the unbounded growth of the memory bank while preserving high-fidelity scene details.

## 4 EXPERIMENT

To validate the effectiveness of our method, we conduct comprehensive experiments on both local and embodied semantic occupancy prediction task with four popular indoor datasets, including Occ-Scannet, EmbodiedOcc-Scannet and their mini version. Details about datasets, implementation, evaluation metrics, additional experiments and visualization are included in our appendix.

### 4.1 MAIN RESULTS

**Local Occupancy Prediction.** As shown in Tab. 1, our method establishes a new state-of-the-art on both the Occ-Scannet-Mini and Occ-Scannet datasets. With standard training protocols, our approach surpasses all baselines. Notably, on Occ-Scannet-Mini, we outperform EmbodiedOcc++ by a margin of +1.52 mIoU. The substantial improvements are particularly evident in large structural

Table 2: **Embodied prediction performance.** We compare our method on EmbodiedOcc-Scannet-Mini and EmbodiedOcc-Scannet datasets against several baselines: SplicingOcc Li et al. (2025a), EmbodiedOcc Wu et al. (2024), RoboOcc Zhang et al. (2025b), and EmbodiedOcc++ Wang et al. (2025a). * denotes aligning local prediction training settings with SplatSSC. † denotes re-training based on the official codes.

| Method | Dataset | IoU | mIoU | ceiling | floor | wall | window | chair | bed | sofa | table | tvs | furniture | objects |
|---|---|---|---|---|---|---|---|---|---|---|---|---|---|---|
| EmbodiedOcc† | EmbodiedOcc-ScanNet-Mini | 50.73 | 40.85 | 19.70 | 43.40 | 39.10 | 28.20 | 45.10 | 61.80 | 53.60 | 40.00 | 29.90 | **56.20** | 32.40 |
| EmbodiedOcc++† | | 50.56 | 40.65 | 13.50 | 42.70 | 38.20 | 28.20 | 43.80 | **63.90** | **54.80** | 39.20 | 36.60 | 55.00 | 31.10 |
| Ours | | **54.39** | **43.96** | **21.90** | **46.50** | **44.00** | **31.70** | **49.40** | 62.30 | 51.50 | **46.00** | **38.40** | 55.10 | **36.80** |
| EmbodiedOcc* | EmbodiedOcc-ScanNet-Mini | 53.53 | 45.01 | **30.10** | 44.80 | 41.60 | 33.50 | 47.00 | 65.20 | 54.30 | 47.40 | 35.90 | 57.80 | 37.40 |
| Ours* | | **57.94** | **48.92** | 29.30 | **50.40** | **45.60** | **36.20** | **51.80** | **66.40** | **57.60** | **49.00** | **48.30** | **61.30** | **42.00** |
| SplicingOcc | EmbodiedOcc-ScanNet | 49.01 | 40.74 | **31.60** | 38.80 | 35.50 | 36.30 | 47.10 | 54.50 | 57.20 | 34.40 | 32.50 | 51.20 | 29.10 |
| EmbodiedOcc | | 51.52 | 42.53 | 22.70 | 44.60 | 37.40 | 38.00 | 50.10 | 56.70 | 59.70 | 35.40 | 38.40 | 52.00 | 32.90 |
| EmbodiedOcc++ | | 52.20 | 43.60 | 27.90 | 43.90 | 38.70 | 40.60 | 49.00 | 57.90 | 59.20 | 36.80 | 37.80 | 53.50 | 34.10 |
| RoboOcc | | 53.30 | 44.05 | 21.94 | 44.57 | 39.54 | 38.48 | 51.28 | 57.04 | 63.09 | 36.70 | **43.05** | 54.42 | 34.38 |
| Ours | | **56.01** | **45.20** | 20.50 | **48.60** | **42.90** | **43.10** | **53.00** | **58.10** | **63.20** | **38.80** | 36.60 | **55.80** | **36.50** |
| EmbodiedOcc* | EmbodiedOcc-ScanNet | 53.53 | 44.90 | 27.50 | 45.60 | 41.00 | 42.60 | 49.10 | 55.70 | 60.50 | **40.60** | 39.20 | 55.20 | 36.80 |
| Ours* | | **56.29** | **47.77** | **27.80** | **50.30** | **44.20** | **45.40** | **53.30** | **60.40** | **63.30** | 39.10 | **44.70** | **57.00** | **40.00** |

categories such as "ceiling", "floor", and "wall", which directly validates the effectiveness of our GGI module in capturing accurate scene geometry. This strong performance seamlessly scales to the larger and more diverse Occ-Scannet dataset, where our method demonstrates consistent gains across all the prior works.

Furthermore, to ensure a direct and fair comparison with methods employing different training schedules, such as SplatSSC, we report results using an optimized learning rate (marked with *). Under this setting, our performance lead extends significantly. On Occ-Scannet-Mini, our method achieves 54.00 mIoU, surpassing SplatSSC by a remarkable +5.13 mIoU. Similarly, on the full OccScannet dataset, we achieve 52.69 mIoU, further demonstrating the significant capabilities and superiority of our proposed framework.

**Embodied Occupancy Prediction.** For the embodied task, our method is evaluated on the EmbodiedOcc-Scannet-Mini and EmbodiedOcc-Scannet datasets, with results presented in Tab. 2. Our approach outperforms all competitors, establishing a new state-of-the-art. On the full EmbodiedOcc-Scannet dataset, for instance, we achieve a +2.69 IoU and +1.15 mIoU improvement over the previous best method. This superior performance is a direct result of our framework's synergistic design: the geometry-guided initialization provides a strong per-frame prior, the globally-aware features ensure spatiotemporal consistency, and the confidence-based fusion efficiently manages the scene representation by pruning redundancies.

To further probe the upper bounds of our method, we also report results using an optimized learning rate (marked with *). Under this setting, the performance gap widens significantly on both datasets. We surpass EmbodiedOcc by +3.91 and +2.87 in mIoU, respectively. The consistent and substantial gains across nearly all semantic categories underscore the robust superiority of our proposed framework for the embodied perception task.

## 4.2 ABLATION STUDIES

We conduct a comprehensive ablation study to validate the contribution of each proposed module. The results for the local and embodied prediction tasks are in Tab. 3 and Tab. 4, respectively.

**Local Occupancy Prediction.** As shown in Tab. 3, each of our proposed components progressively improves performance over the EmbodiedOcc baseline (Row 1). Our GGI module alone provides the most significant initial boost (+1.70 mIoU), confirming the critical impact of an effective, parameter-free initialization strategy (Row 2). Building upon this, both the learnable offsets (Row 3) and the PAGR (Row 4) contribute further gains by enabling scene-adaptive refinement and enhancing spatial awareness, respectively. Finally, our full model (Row 5) achieves the best performance, demonstrating a clear synergistic effect and validating the effectiveness of our overall design.

**Embodied Occupancy Prediction.** We ablate our components for the embodied task in Tab. 4, introducing two metrics to evaluate efficiency: Average Gaussians per Scene (AGS) and Gaussian

Table 3: Component ablation on local occupancy prediction task. Experiments are conducted on Occ-Scannet-Mini dataset.

| # | Components | | | Metrics | |
|---|---|---|---|---|---|
| | GGI | learnable offsets | PAGR | IoU | mIoU |
| 1 | | | | 53.72 | 46.05 |
| 2 | ✓ | | | 55.19 | 47.75 |
| 3 | ✓ | ✓ | | 56.47 | 49.08 |
| 4 | ✓ | | ✓ | 56.80 | 48.75 |
| 5 | ✓ | ✓ | ✓ | **57.67** | **49.72** |

Table 4: Component ablation on embodied occupancy prediction task. Experiments are conducted on EmbodiedOcc-Scannet dataset. "Dense" denotes setting Gaussian initializing interval to 0.08m.

| # | Components | | | | Metrics | | | |
|---|---|---|---|---|---|---|---|---|
| | Dense | GGI | CBFP | PAGR | IoU | mIoU | AGS | GGR |
| 1 | | | | | 53.53 | 44.90 | 16,301 | 1.0 |
| 2 | ✓ | | | | 55.11 | 46.49 | 132,522 | 8.13 |
| 3 | | ✓ | | | 56.22 | 47.13 | 63,951 | 3.92 |
| 4 | | ✓ | ✓ | | 56.23 | 47.29 | 53,081 | 3.25 |
| 5 | | ✓ | ✓ | ✓ | **56.29** | **47.77** | 52,881 | 3.24 |

Growth Rate (GGR). Our GGI module (Row 3) substantially outperforms the standard EmbodiedOcc* baseline by +2.23 mIoU. More critically, it surpasses a much denser baseline (Row 2) by +0.64 mIoU while using only 48% of the Gaussian primitives, clearly demonstrating that our performance gains stem from efficient placement, not just increased density. Building on this, the CBFP module (Row 4) further improves efficiency, reducing the average primitive count by 17% while also yielding a +0.16 mIoU gain. This confirms its ability to prune redundancy without sacrificing key information. Finally, the addition of the PAGR module (Row 5, our full model) provides another +0.48 mIoU performance boost even on top of an already strong model, underscoring the critical role of global spatial information for consistent and accurate embodied perception.

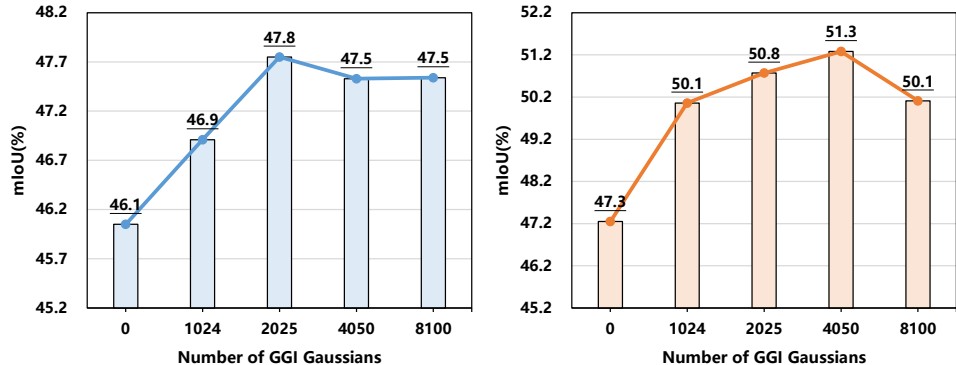

Figure 4: Sensitivity to the Number of Initialized Primitives.

**Sensitivity to Initialization Strategy.** We analyze the sensitivity of our model to the number of geometry-guided Gaussian primitives, with results shown in Fig. 4. The experiment varies the proportion of guided primitives versus randomly initialized ones while keeping the total count fixed, using both estimated and ground-truth (GT) depth.

The results yield three key insights. First, any amount of geometric guidance significantly outperforms the purely random baseline (0 guided primitives), confirming the robustness and efficiency of our core strategy. Second, performance follows a non-linear, concave trend, peaking at 2,025 and 4,050 guided primitives for estimated and GT depth, respectively. This suggests an optimal trade-off between the strength of the geometric prior and the stochastic exploration afforded by random primitives. The peak's shift with higher-quality GT depth further supports this trade-off hypothesis. Finally, our explicit initialization strategy dramatically improves the utilization of high-quality depth information. While the baseline's implicit depth usage yields only a +1.2 mIoU gain with GT depth, our method leverages it for a far greater +5.2 mIoU boost, demonstrating a more effective translation of geometric priors into performance.

**Analysis of the CBFP Module.** In Tab. 5, We ablate the key designs of our fusion module, on the weighting strategy and the grid size. The results show that our proposed confidence-based weighting (Row 4) is critical. While simpler strategies like mean (Row 2) or opacity-based (Row 3) fusion reduce primitive count at the cost of accuracy, our method simultaneously reduces the GGR from 3.87 to 3.24 and improves performance by +0.20 mIoU.

Furthermore, the grid size analysis (Row 4-6) reveals a clear trade-off between pruning efficiency and accuracy. A small grid (0.02m) is ineffective at reducing primitives, while a large grid (0.08m) significantly harms performance (-1.16 mIoU) due to over-fusion. Our chosen 0.04m grid size strikes the optimal balance, achieving substantial efficiency gains while improving prediction accuracy, and is used as our final configuration.

Table 5: Analysis of the Gaussian Primitive Fusion Module.

| Index | Fusion Mode | | | Grid Size | | | Metrics | | | |
|---|---|---|---|---|---|---|---|---|---|---|
| | Avg. | Opa. | Conf. | 0.02 | 0.04 | 0.08 | IoU | mIoU | AGS | GGR |
| 1 | | | | | | | 56.22 | 47.57 | 63,148 | 3.87 |
| 2 | ✓ | | | | ✓ | | 56.23 | 47.29 | 52,845 | 3.26 |
| 3 | | ✓ | | | ✓ | | 56.17 | 47.11 | 52,779 | 3.24 |
| 4 | | | ✓ | | ✓ | | 56.29 | **47.77** | 52,881 | 3.24 |
| 5 | | | ✓ | ✓ | | | **56.48** | 47.58 | 60,920 | 3.73 |
| 6 | | | ✓ | | | ✓ | 55.70 | 46.61 | 37,025 | 2.27 |

**Visualization of Local Occupancy Prediction.** Qualitative results in Fig. 5 demonstrate the superiority of our method over the EmbodiedOcc baseline. Our model excels at reconstructing both large-scale scene geometry, producing more complete and detailed surfaces for structures like walls and floors, and individual objects, rendering them with more precise contours while mitigating common artifacts like holes and fragmented edges. These visual improvements provide intuitive validation for our approach. By leveraging a geometry-guided initialization and globally-aware features, our model constructs a superior geometric and semantic prior, which directly translates to higher structural integrity and finer detail accuracy in the final prediction.

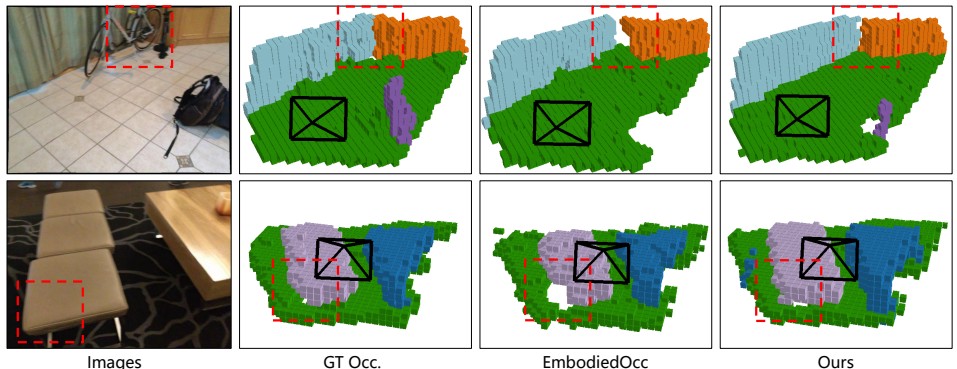

Figure 5: Visualization of local occupancy prediction.

## 5 CONCLUSION

In this work, we address the critical challenge of inefficient initialization in Gaussian-based embodied occupancy prediction. We propose a novel framework that leverages geometric priors, recovered from monocular images, to manage the entire lifecycle of Gaussian primitives. Our approach discards the naive uniform initialization of prior work and instead employs a geometry-guided initialization module that places primitives on potentially occupied regions, directly addressing the suboptimal allocation problem. Furthermore, we introduce a Position-Aware Scene Update and Pruning pipeline that integrates global position encoding for spatiotemporal consistency and confidence-based fusion to efficiently prune redundancies. Extensive experiments demonstrate that our method establishes a new state-of-the-art on four mainstream benchmarks, providing an effective paradigm for building more efficient and robust vision-only 3D scene understanding systems.

## 6 ETHICS STATEMENT

This work adheres to the ICLR Code of Ethics. In this study, no human subjects or animal experimentation was involved. All datasets used, including Occ-Scannet, EmbodiedOcc-Scannet and their mini version, were sourced in compliance with relevant usage guidelines, ensuring no violation of privacy. We have taken care to avoid any biases or discriminatory outcomes in our research process. No personally identifiable information was used, and no experiments were conducted that could raise privacy or security concerns. We are committed to maintaining transparency and integrity throughout the research process.

## 7 REPRODUCIBILITY STATEMENT

To ensure reproducibility, we have detailed our experimental setup within the paper, including training procedures, model configurations, hardware specifics, and a full description of our Geometry-Guided Embodied Occupancy Perception Pipeline.

Furthermore, all datasets used in this study, including Occ-Scannet, EmbodiedOcc-Scannet, and their mini versions, are publicly available. The complete code for this paper will be made publicly available on GitHub upon acceptance of the manuscript. We believe these measures provide sufficient detail for other researchers to reproduce our work and build upon it.

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

## A APPENDIX

### A.1 EXPERIMENTAL SETUP

**Datasets.** The proposed method is evaluated on the Occ-ScanNet Yu et al. (2024) and EmbodiedOcc-ScanNet Wu et al. (2024) datasets. Occ-ScanNet provides monocular images paired with voxel-level semantic occupancy annotations, with its training and validation sets containing 45,755 and 19,764 samples, respectively. The ground truth occupancy for each frame is represented by a $60 \times 60 \times 36$ voxel grid, corresponding to a $4.8m \times 4.8m \times 2.88m$ space in front of the camera with a voxel resolution of $0.08m$. Its semantic annotations cover 12 distinct classes, comprising one 'free space' class and 11 meaningful object and structural categories, which include architectural elements (ceiling, floor, wall, window), furniture (chair, bed, sofa, table), electronics (TVs), and a general 'objects' category. In addition to the full dataset, Occ-ScanNet also offers a smaller subset, Occ-ScanNet-mini, with 5,504 and 2,376 samples in its training and validation splits, respectively. EmbodiedOcc-ScanNet is built upon Occ-ScanNet and is divided into 537 training scenes and 137 validation scenes. Each scene comprises 30 posed color images and their corresponding occupancy annotations. It also features a mini version, EmbodiedOcc-ScanNet-mini, which contains 64 training and 16 validation scenes. The semantic classes for the ground truth occupancy are identical to those in Occ-ScanNet. The resolution for global occupancy, however, varies with the scene dimensions and is calculated as $(l_x \times l_y \times l_z)/0.08m$, where $l_x, l_y, l_z$ represent the spatial extent of the scene in world coordinates.

**Tasks.** Following the setup in EmbodiedOcc Wu et al. (2024), the proposed method is evaluated on two distinct tasks: local occupancy prediction and embodied occupancy prediction. For the local occupancy prediction task, the established paradigm from prior works Yu et al. (2024); Wu et al. (2024) is followed, which uses a single monocular image to predict occupancy within the camera's view frustum. For the more challenging embodied occupancy prediction task, the proposed method sequentially processes visual inputs, utilizing the local occupancy prediction from the current frame to continuously update the global occupancy prediction.

**Metrics.** Adhering to previous works Yu et al. (2024); Wu et al. (2024), the evaluation metrics include two key performance indicators: Scene Completion Intersection over Union (IoU) and mean Intersection over Union (mIoU) for semantic understanding. The Scene Completion IoU serves as a comprehensive metric for assessing overall occupancy prediction accuracy, while the per-class mIoU provides detailed insights into the model's performance across different semantic categories. For local occupancy prediction, the evaluation strictly follows the protocol from ISO Yu et al. (2024), computing these metrics within the camera's view frustum. For embodied occupancy prediction, the evaluation scope is expanded to the global occupancy of each scene, with a focus on the regions observed throughout the entire 30-frame sequence.

### A.2 IMPLEMENTATION DETAILS

**Local Occupancy Prediction.** Following prior works Yu et al. (2024); Wu et al. (2024), a pre-trained EfficientNet-B7 Tan & Le (2019) is employed as the image encoder to extract multi-scale semantic features. For the initialization of Gaussian primitives, a fine-tuned and frozen DepthAnythingV2 Yang et al. (2024) model is utilized to predict the depth map and a frozen MoGe-2 Wang

et al. (2025b) model is utilized to predict the normal map. Regarding hyperparameters, the monocular image resolution is set to 480×640. The total number of Gaussian primitives is 16,200, comprising 2,205 geometry-guided primitives and 14,175 randomly initialized primitives. The scale of each Gaussian primitive is capped at 0.08m. For the training setup, the AdamW Loshchilov & Hutter (2017) optimizer is used with a weight decay of 0.01. The learning rate is warmed up to a maximum of 2e-4 over the first 1000 iterations and subsequently decreased following a cosine annealing schedule. The model is trained on 8 RTX A6000 GPUs for 10 epochs on the Occ-ScanNet dataset and 20 epochs on the Occ-ScanNet-mini subset.

**Embodied Occupancy Prediction.** The model is fine-tuned for the embodied occupancy prediction task, starting from the weights pre-trained on local prediction. Specifically, for each frame, the model generates 2,025 new geometry-guided Gaussian primitives from the current observation. These new primitives, along with those from the existing memory bank, are used to update the global occupancy prediction, thereby achieving a comprehensive reconstruction of the scene. The model is trained for 5 epochs on the EmbodiedOcc-ScanNet dataset using 8 RTX A6000 GPUs and 20 epochs on the EmbodiedOcc-ScanNet-Mini dataset using 4 RTX A6000 GPUs. All other settings remain consistent with the training process for local occupancy prediction.

## A.3 ADDITIONAL EXPERIMENTS

**Ablation on PAGR process.** We ablate our PAGR process to validate its effectiveness, with results shown in Fig. 6. We evaluate fusing the positional encoding with: (a) only GPAF module, (b) only GPAE module, and (c) the PAGR process. The results demonstrate that while each fusion path individually improves performance over the baseline, applying the encoding to both branches yields a significant synergistic effect. This fusion boosts performance by an additional +0.4 and +0.3 mIoU on the local and embodied tasks respectively, confirming the optimality of our design.

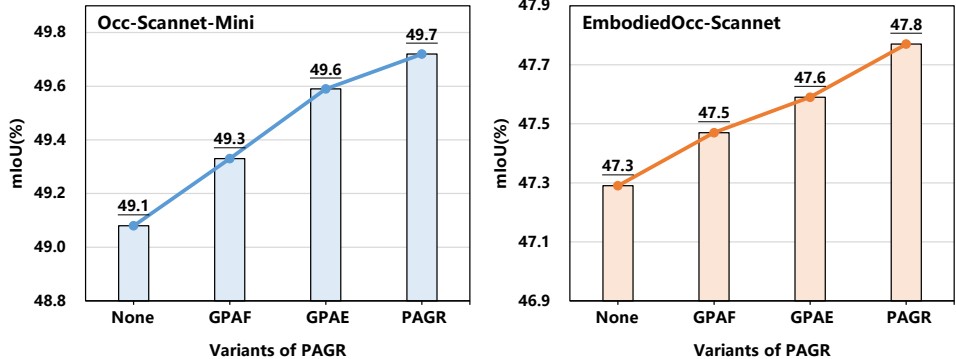

Figure 6: Ablation on PAGR module.

## A.4 ADDITION QUALITATIVE ANALYSIS

**Qualitative Analysis of Embodied Occupancy Prediction.** The qualitative results in Fig. 7 highlight our method's superiority for the embodied task. Our framework demonstrates high structural integrity and consistency, which is attributable to the synergy between our geometry-guided initialization that ensures comprehensive surface coverage and our globally-aware features that provide a stable spatial reference across frames. Furthermore, our model excels at progressive refinement by integrating multi-view observations. This is enabled by our confidence-based fusion mechanism, which effectively suppresses transient estimation errors from challenging viewpoints (*e.g.*, occlusions), thereby enhancing the final reconstruction's robustness and accuracy while pruning redundancies.

## A.5 USE OF LARGE LANGUAGE MODELS

Large Language Models (LLMs) were used to aid in the writing and polishing of the manuscript. Specifically, we used an LLM to assist in refining the language, improving readability, and ensuring

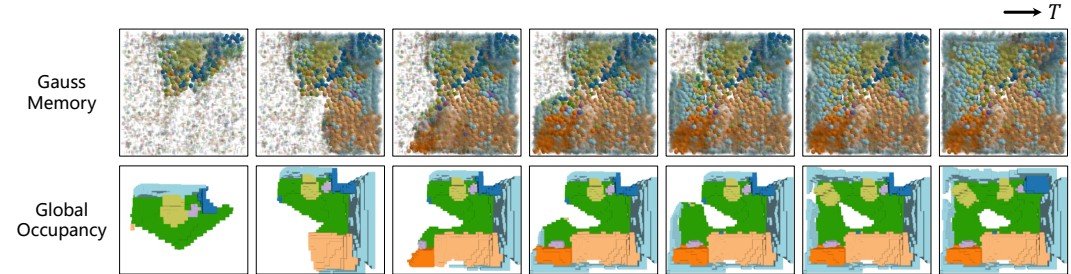

Figure 7: Visualization of embodied occupancy prediction.

clarity in various sections of the paper. The model helped with tasks such as sentence rephrasing, grammar checking, and enhancing the overall flow of the text.

It is important to note that the LLM's role was strictly limited to improving the linguistic quality of the paper. The model was not involved in any part of the scientific process, including ideation, research methodology, experimental design, or data analysis. All research concepts, ideas, and analyses were developed and conducted solely by the authors, who take full responsibility for the entire content of the manuscript. We have ensured that all LLM-assisted text adheres to ethical guidelines and does not constitute plagiarism or scientific misconduct.

