# OpenReview forum: "G$^2$-Occ: Geometry-Guided Gaussian Primitives for Embodied Semantic Occupancy Prediction"
_ICLR.cc/2026/Conference — Submitted to ICLR 2026_

### Official Review · Reviewer_4k2W · 2025-10-16

**Soundness:** 2
**Presentation:** 2
**Contribution:** 3
**Rating:** 4
**Confidence:** 5

**Summary:**

This paper proposes G2-OCC, a vision-only framework for embodied semantic occupancy prediction model.
The core design of this work is a Geometry-Guided Initialization module that leverages the recovered geometric priors to generate Gaussian primitives within potentially occupied regions of the scene, improving the efficiency.
To handle the embodied Gaussian memory update, this paper proposes a Position-Aware Scene Update and Pruning pipeline.
Experiments show that the proposed Geometry-Guided Initialization module is effective.

**Strengths:**

1. The problems this paper targets are clear, and the proposed solution (Geometry-Guided Initialization module) seems effective according to the experimental results.
2. The proposed Geometry-Guided Initialization module is intuitive and easy to understand.

**Weaknesses:**

1. I noticed that you compared with SplatSSC in the experimental section. I'm curious about the differences between your proposed Geometry-Guided Initialization module and SplatSSC's depth-guided initialization strategy. And what are your advantages?
In addition, Table 1 presents the experimental results of several methods under the training setting of SplatSSC, which shows a significant performance improvement compared to the normal version (like Ours). I am wondering if only the learning rate was modified during this process? Does this imply that the performance of current methods is highly dependent on the setting of experimental parameters? I believe there should be more discussions on this aspect.
2. In the Position-Aware Gaussian Refinement, you state that updating Gaussians in the global absolute coordinate system is more optimal. From my understanding, these coordinate systems only have a certain transformation relationship. Does requiring all Gaussians to be updated in a fixed world coordinate system introduce additional bias priors into the model? In a truly embodied setting, if we do not know the size of the environment to be explored in advance, the predefinition of the world coordinate system is unreasonable. Moreover, from the results in Table 4, adding the Position-Aware Gaussian Refinement does not bring about a significant performance improvement.
3. Similarly, the experimental results in Table 5 fail to effectively prove the advantages of Confidence-Based Fusion and Pruning. Moreover, since a more refined paradigm has been incorporated, I believe it is necessary to conduct a more comprehensive comparison between G2-OCC and the baselines, such as in terms of running time, peak memory usage, training time, and so on.

Minor Weaknesses:
1. Some citations (like Line48 & Line49) are improper.
2. I think abbreviating many small modules used in the article in a similar way greatly affects the reading experience. For those who are not familiar with the names of these modules you proposed, such abbreviations require readers to repeatedly check back to confirm which words they specifically refer to (like CBFP, PAGR).
3. As mentioned in the abstract, this work has conducted sufficient experiments on four popular indoor semantic occupancy prediction benchmarks, but this seems an overclaim regarding the workload. Occ-ScanNet and its miniset actually belong to the same benchmark, as their data distributions are highly similar.

Please answer the above questions carefully and provide more thorough discussion and comparison.

**Questions:**

Please refer to the Weaknesses.

---

### Official Review · Reviewer_vu48 · 2025-10-23

**Soundness:** 2
**Presentation:** 2
**Contribution:** 2
**Rating:** 4
**Confidence:** 4

**Summary:**

The paper addresses the task of monocular 3D occupancy prediction. The authors propose a geometry-guided initialization framework and a position-aware update–pruning pipeline for Gaussian primitives. The proposed components effectively leverage geometric priors and spatiotemporal consistency to improve both initialization efficiency and scene representation quality. Extensive experiments conducted on four indoor 3D semantic occupancy benchmarks show that the method achieves new state-of-the-art results, significantly outperforming previous approaches.

**Strengths:**

1. The proposed method achieved good performance on several indoor benchmarks.
2. The ablation study is comprehensive and the figure is clear

**Weaknesses:**

1. The technical novelty is somewhat limited. The overall pipeline is quite similar to EmbodiedOcc++, as both adopt geometry-guided Gaussians and maintain a Gaussian memory.
2. Most previous monocular occupancy prediction methods operate in an online manner. As shown in Fig. 1, the proposed method appears to rely on global image information; How is it fundamentally different from reconstruction? Maybe I misunderstood the current setting?
3. Several prior works, such as SplatSSC, employ depth-based initialization for Gaussians instead of random initialization. It would be important to clarify how the proposed initialization differs from these methods and what specific problem it solved.

**Questions:**

1. What is the latency of the proposed method?
2. How much does the quality of the initial depth estimation affect the overall performance of the model?

---

### Official Review · Reviewer_8w67 · 2025-10-31

**Soundness:** 2
**Presentation:** 1
**Contribution:** 2
**Rating:** 2
**Confidence:** 5

**Summary:**

This paper introduces $G^2-Occ$, a framework for embodied semantic occupancy prediction using Geometry-Guided Gaussian primitives. By leveraging depth and normal priors, $G^2-Occ$ initializes sparse Gaussian primitives in occupied regions and refines them through position-aware updates and confidence-based pruning. Experiments on Occ-ScanNet and EmbodiedOcc-ScanNet demonstrate the method’s effectiveness, achieving state-of-the-art performance in both local and embodied occupancy prediction tasks.

**Strengths:**

The paper demonstrates strong experimental results, achieving state-of-the-art performance on the embodied occupancy prediction task.

**Weaknesses:**

1. The paper's novelty is limited. The core contributions, including the Geometry-Guided Initialization (GGI) module and the Position-Encoding module, are incremental. The GGI module closely resembles prior work in GaussianFormer-2 [1], which also leverages depth priors to initialize Gaussians in occupied regions, while the Position-Encoding module is relatively simple in design. Both innovations lack significant originality.

2. The experimental comparison with EmbodiedOcc [2] is unfair and fails to validate the effectiveness of the proposed innovations. As shown in the supplementary material, $G^2-Occ$ employs Moge-v2 [3], a much stronger visual geometry model, to provide geometric priors. This raises doubts about whether the performance gains are due to the GGI module. Furthermore, $G^2-Occ$ relies on only a small number (2,205) of geometry-guided primitives while using a large number (14,175) of randomly initialized primitives, which questions the motivation behind the GGI module.

3. The paper lacks sufficient visualization of Gaussian distributions to validate the effectiveness of the GGI module, such as showing more Gaussians distributed in occupied regions.

4. Additional experiments are needed to analyze the efficiency impact of the GGI module, considering that the usage of Moge-v2 may introduce extra computational overhead.

5. The paper needs to be revised for better readability. Excessive use of abbreviations for proposed modules, such as Geometry-Guided Initialization (GGI), Position-Aware Gaussian Refinement (PAGR), Global Position-Aware Feature Refinement (GPAF), Global Position-Aware Encoder (GPAE), and Confidence-Based Fusion and Pruning (CBFP), makes it difficult to clearly identify the core innovations.

**Minor issues**:
1. The distinction between "Gauss" and "Gaussian" needs to be clarified.
2. The meaning of the y-axis in Figure 3 (left) should be explicitly stated.
3. The meanings of the left and right plots in Figure 4 are unclear.

**Reference**

[1] GaussianFormer-2: Probabilistic Gaussian Superposition for Efficient 3D Occupancy Prediction.

[2] EmbodiedOcc: Embodied 3D Occupancy Prediction for Vision-based Online Scene Understanding.

[3] Moge-2: Accurate monocular geometry with metric scale and sharp details.

**Questions:**

See weaknesses.

---

### Meta-Review · Area_Chair_uqNm · 2025-12-02

**Summary:**

The reviewers raised substantial concerns regarding the limited novelty of the proposed method, the unclear distinction from prior work, and the insufficient experimental evidence supporting the claimed contributions. Multiple reviewers pointed out that key modules, such as the Geometry-Guided Initialization and Position-Aware Refinement, appear incremental and closely related to existing methods including GaussianFormer-2, EmbodiedOcc++, and SplatSSC.

Concerns were also expressed about the fairness of comparisons, reliance on stronger depth priors, insufficient visualization of Gaussian distributions, limited analysis of computational efficiency, and unclear or incomplete explanations throughout the paper. Although the method shows strong empirical performance, the unresolved novelty and validation issues informed the suggested decision.

**Reviewer Concerns:**

Since no author response was provided, none of the reviewers’ concerns were addressed, including:

- Novelty concerns regarding similarity to GaussianFormer-2, EmbodiedOcc++, and SplatSSC.
- Unfair or unclear experimental comparisons, particularly the reliance on strong geometry models such as Moge-v2.
- Missing visualizations to demonstrate the effectiveness of the proposed initialization strategy.
- Lack of efficiency analysis, such as latency, computational cost, and memory usage.
- Clarity issues, including abbreviations, module naming, and ambiguous figure descriptions.
- Questions about coordinate system choices, bias introduction, and embodied setting assumptions.
- Requests for deeper discussion on hyperparameter sensitivity and the dependence on depth estimation quality.

**Reviewer Scores:**

Because the authors did not provide a rebuttal and therefore did not clarify or correct any reviewer misunderstandings, it is reasonable to expect that reviewer scores would remain unchanged for all reviewers.

- Reviewer 8w67 (Rating: 2 – reject)
- Reviewer vu48 (Rating: 4 – marginally below acceptance)
- Reviewer 4k2W (Rating: 4 – marginally below acceptance)

---

### Decision · Program_Chairs · 2026-01-26

Reject